# FORMULAREASONING: A DATASET FOR FORMULA-BASED NUMERICAL REASONING

## ABSTRACT

The application of formulas is a fundamental ability of humans when addressing numerical reasoning problems. However, existing numerical reasoning datasets seldom indicate explicitly the formulas employed during the reasoning steps. To bridge this gap, we construct a dataset for formula-based numerical reasoning called FormulaReasoning, which consists of 5,420 reasoning-based questions. We employ it to conduct evaluations of LLMs with size ranging from 7B to over 100B parameters utilizing zero-shot and few-shot chain-of-thought methods, and we further explore using retrieval-augmented LLMs provided with an external formula database associated with our dataset. We also experiment with supervised methods where we divide the reasoning process into formula generation, parameter extraction, and numerical calculation, and perform data augmentation. Our empirical findings underscore the significant potential for improvement in existing models when applied to our challenging, formula-driven FormulaReasoning.

## 1 INTRODUCTION

Numerical reasoning constitutes one of the significant forms within natural language reasoning (Frieder et al., 2023). The study of numerical reasoning has seen substantial progress in recent years, largely driven by the development of LLMs (OpenAI, 2023; Touvron et al., 2023; Li et al., 2023c) and specialized datasets (Wang et al., 2017; Dua et al., 2019; Amini et al., 2019; Cobbe et al., 2021a). Current datasets for numerical reasoning typically include simple, commonsense numerical questions that do not reflect the complexity of real-world problems. These datasets have not fully addressed the interpretability issue in numerical reasoning, as they often rely on implicit commonsense knowledge without explicit guidance knowledge during the reasoning process. This issue becomes particularly evident when LLMs meet hallucination (Frieder et al., 2023; Bang et al., 2023). Consequently, one might naturally ask *"What knowledge could be used to guide numerical reasoning process?"*. Formulas exactly represent such knowledge that has been largely overlooked in previous research but is frequently utilized in real-life applications.

Take a question from the GSM8K (Cobbe et al., 2021a) as an example: "A robe takes 2 bolts of blue fiber and half that much white fiber. How many bolts in total does it take?". This example only requires the use of implicit *commonsense mathematical knowledge* to solve without domain-specific formula. However, in our FormulaReasoning dataset, we require *domain-specific formulas* to guide the numerical reasoning process, such as the formula used to calculate the heat absorption of an object.

Recently, Liu et al., 2023 constructed two formula-based datasets, Math23K-F and MAWPS-F. However, the formulas in these datasets primarily consist of commonsense formulas (such as total_amount = unit_amount × total_number), and only 33.5% and 38.4% of the questions in these datasets, respectively, require the use of formulas.

To fill this gap, we constructed a dataset for numerical reasoning that requires the use of formulas called FormulaReasoning. We annotated formulas for each question in FormulaReasoning. An example of FormulaReasoning is shown in Figure 1.[1] The formula-based feature makes Formula-Reasoning a more challenging dataset for developing systems that can tackle real-world numerical

---

[1]Please note that FormulaReasoning is originally in Chinese. For the convenience of understanding, we translated Chinese into English in all the examples presented in this paper.

**Question**

> There is a electric water heater, after 50kg of water is loaded into its tank, the water is heated from 20℃ to 60℃ by electricity. It is known that the specific heat capacity of water is C_water = 4.2×10^3J/(kg*℃).
> Q: If the total electrical energy consumed during the heating process is 1×10^7J, what is the thermal efficiency of the water heater?

**Explanation (Reasoning Steps)**

> Calculating the degree of temperature increase in water: [Degree of water temperature increase] = [Final temperature] − [Initial temperature] = 60 ℃ − 20 ℃ = 40 ℃. The degree of water temperature increase = 40 ℃.
> The heat absorbed by water is given by: [Heat absorbed by water] = [Mass of water] ∗ [Specific heat capacity of water] ∗ [Degree of water temperature increase] = 50 kg ∗ 4.2 ∗ 10^3 J/(kg·℃) ∗ 40 ℃ = 8400000 J. The heat absorbed by water = 8400000 J.
> The thermal efficiency of the water heater can be obtained from: [Thermal efficiency of the water heater] = [Heat absorbed by water] / [Total electrical energy consumed] ∗ 100% = 8400000 J / (1 ∗ 10^7 J) ∗ 100% = 84%. The thermal efficiency of the water heater = 84%.
> Answer = 84%

**Parameter Table**

| Parameter | Symbol | Value | Unit |
|---|---|---|---|
| Degree of water temperature increase | $\Delta t$ | 40 | ℃ |
| Final temperature | $t_{final}$ | 20 | ℃ |
| ... | ... | ... | ... |
| Heat absorbed by water | $Q_{absorbed}$ | 8400000 | J |
| Mass of water | $m_{water}$ | 50 | kg |

Figure 1: An example taken from FormulaReasoning. Numerical values (including units) given in the question and obtained from intermediate steps are highlighted in red and purple, respectively. Formulas and their elements are in blue.

reasoning problems. Indeed, in fields such as mathematics and physics, formulas serve as an important vessel for representing domain knowledge. However, existing datasets scarcely consider explicit incorporation of formulas into numerical reasoning.

Table 1: Statistics of Math23-F, MAWPS-F, GSM8K, MATH and our FormulaReasoning.

| Dataset | Math23K-F | MAWPS-F | GSM8K | MATH | FormulaReasoning |
|---|---|---|---|---|---|
| # questions | 23,162 | 2,373 | 8,792 | 12,500 | 5,420 |
| # formulas (and variants) | 51 (131) | 18 (46) | 0 (0) | 0 (0) | 272 (824) |
| # questions requiring formula (proportion) | 7,750 (33.46%) | 911 (38.39%) | N/A | N/A | 5,420 (100%) |
| Avg. # reasoning steps | 1.16 | 1.01 | 3.59 | Not Provided | 2.37 |

We collected questions requiring formula-based numerical reasoning from Chinese junior high school physics examinations. With the *combined efforts of manual annotation and assistance from LLMs*, we annotated each question with an explanation text, a final answer, and a set of relevant formulas (including formula structures, parameter names, symbols, numerical values, and units) and built a *consolidated formula database*. The formula database functions as an external knowledge base, which can be used to evaluate retrieval-based/augmented systems. In Table 1, we compare FormulaReasoning with two existing formula-based datasets and the well-known GSM8K and MATH (Hendrycks et al., 2021). In comparison to Math23K-F and MAWPS-F, FormulaReasoning contains *a larger number of formulas* (272), whereas the other two datasets contain 51 and 18 formulas. Additionally, all questions in FormulaReasoning require the use of formulas. The *higher average number of reasoning steps* (2.37 vs. 1.16/1.01) implies that FormulaReasoning is more challenging and better suited for evaluating existing models as a multi-step formula-based reasoning task.

We used FormulaReasoning to evaluate LLMs ranging from 7B to >100B parameters, as well as fine-tuned models such as Qwen-1.8B (Bai et al., 2023) and ChatGLM3-6B (Zeng et al., 2022) with a proposed Chain-of-Thought supervised fine-tuned method and a data augmentation method. We also trained an encoder for formula retrieval and experimented with retrieval-augmented generative

models. Our empirical findings show that the best existing models only achieve an accuracy of around 84%, lagging behind an accuracy 92% of humans, indicating that there is still significant room for exploration in formula-based numerical reasoning.

Our contributions are summarized as follows:

- We construct a formula-based numerical reasoning dataset FormulaReasoning, with fine-grained annotations for each question. As a formular knowledge-guided numerical reasoning dataset, it can be applied to tasks involving trustworthy and verifiable reasoning.
- We conduct evaluations on LLMs of various sizes, supervised fine-tuned models, and retrieval-augmented generative models. The experimental results establish a strong baseline for future research and also indicate that the task remains unresolved.

The dataset and code is currently available on anonymous GitHub `https://anonymous.4open.science/r/FormulaReasoning`.

## 2 RELATED WORK

### 2.1 NUMERICAL REASONING DATASETS

Numerical reasoning is one of the fundamental capabilities of natural language reasoning. The study of numerical reasoning in natural language has existed for several years. Numerous datasets, such as DROP (Dua et al., 2019), GSM8K (Cobbe et al., 2021b), TSQA (Li et al., 2021) and MATH (Hendrycks et al., 2021), have introduced natural language numerical reasoning. Another line of research focusing on numerical reasoning in natural language is math word problem (MWP). MWP tasks typically provide a short passage (i.e., a question) and require the generation of an arithmetic expression that can compute an answer. Representative datasets include MAWPS (Koncel-Kedziorski et al., 2016), Math23K (Wang et al., 2017), MathQA (Amini et al., 2019), etc. Several works focus on specialized domains where some of the questions in their datasets require numerical reasoning. Examples include GeoSQA (Huang et al., 2019), which focuses on the geography domain, the STEM (Drori et al., 2023) dataset and the ScienceQA (Lu et al., 2022) which covers multiple disciplines in science and technology. The distinguishing feature of our FormulaReasoning is that the numerical reasoning questions within these datasets lack explicitly labeled formulas.

The recently introduced datasets (Liu et al., 2023) Math23K-F and MAWPS-F require formulas for only 33.5% and 38.4% of the questions, respectively, and the formulas within these datasets are all simple commonsense formulas (e.g., total_cost = unit_cost × total_number). By contrast, our FormulaReasoning dataset collects questions from junior high school physics examinations, with every question accompanied by formulas. In addition, we also annotated a *formula database* for FormulaReasoning that can serve as an external knowledge base, used to assess retrieval-augmented systems.

### 2.2 NUMERICAL REASONING METHODS

The methods for solving numerical reasoning have evolved from statistical approaches (Hosseini et al., 2014; Kushman et al., 2014) to those based on rules and templates (Shi et al., 2015; Wang et al., 2019) and further to methods based on deep learning models (Gupta et al., 2019; Chen et al., 2022; Kim et al., 2022; Li et al., 2023a). In the past two years, with the rapid development of LLMs, LLMs have demonstrated strong capabilities in resolving numerical reasoning questions. Consequently, several methods aimed at enhancing the reasoning abilities of LLMs have been proposed, including the notable Chain of Thoughts (CoTs) method (Wei et al., 2022), along with many subsequent variant approaches (Kojima et al., 2022; Wang et al., 2022; Zhou et al., 2022; Li et al., 2023b).

We established representative existing methods as baselines for FormulaReasoning, including zero/few-shot CoTs prompting methods to LLMs ranging from 7B to over 100B parameters. We trained a specialized formula retriever for retrieving formulas and explored retrieval-enhanced numerical reasoning. We also divided the reasoning process into formula generation, parameter extraction, and numerical calculation, and used data augmentation to enhance fine-tuned models with fewer than 7B parameters.

## 3 DATASET CONSTRUCTION

We collected raw questions from Chinese junior high school physics examinations from 2015 to the present. We had a total of five postgraduate volunteer students, and they all hold a bachelor's degree in science and engineering. We then annotated the reasoning steps and corresponding formulas for each question. This process involved a combination of manual annotation and the assistance of LLMs to improve the efficiency of annotation. Each question is associated with an explanation of the reasoning steps in natural language with a symbolic representation of the reasoning steps using formulas, including the values and units for all the parameters within the formulas. Finally, we compiled all the formulas and we merged those expressing the same meaning to create a formula database. We describe this process to construct FormulaReasoning in detail below.

### 3.1 PREPROCESSING

We crawled 18,433 junior high school physics examination questions in China from 2015 to the present from public sources, including only those with free-text answers and excluding multiple-choice and true/false questions. Each raw question contains a *question text* and an *explanation text that includes the reasoning steps*. We eliminated questions requiring diagrams.

Subsequently, we filtered the questions by assessing the presence of numerical values within the explanation and confirming that the final answer was numerical. Utilizing a regular expression-based approach, we extracted the *final numerical answer*, including its unit, from the explanation. We found that for 487 questions, the regular expressions did not return results, so we manually annotated the positions of their answers in the text explanations. Following the preprocessing phase, we compiled an initial dataset comprising 6,306 questions.

Table 2: Original explanation and explanation with normalized formulas (highlighted in blue).

---

**Original explanation.**
The change in water temperature is 60 - 20 = 40 °C. Therefore, the heat absorbed by the water is Q_{absorbed}=50 kg $\times$ 4.2 $\times 10^3$ J/(kg·°C) $\times$ 40 °C = 8.4 $\times 10^6$ J. Given that the total electrical energy consumed in the heating process is $1 \times 10^7$ J, the thermal efficiency of the water heater can be calculated using the formula for the efficiency of a heat engine: $\eta$ = Q_{absorbed}}/W_{total}$\times$100% = $(8.4 \times 10^6$ J)/$(1.0 \times 10^7$ J)$\times$100% = 84%. Answer: If it is known that the total electrical energy consumed during the heating process is $1 \times 10^7$, the thermal efficiency of the water heater is 84%.

**Explanation with normalized formulas.**
1. Calculating the temperature increase in water: [Degree of water temperature increase] = [Final temperature] - [Initial temperature] = 60 °C - 20 °C = 40 °C. The degree of water temperature increase = 40 °C.
2. Calculating the heat absorbed by water: [Heat absorbed by water] = [Mass of water] $\times$ [Specific heat capacity of water] $\times$ [Degree of water temperature increase] = 50 kg $\times$ 4.2 $\times 10^3$ J/(kg·°C) $\times$ 40 °C = 8400000 J. The heat absorbed by water = 8400000 J.
3. The thermal efficiency of the water heater can be obtained from: [Thermal efficiency of the water heater] = [Heat absorbed by water] / [Total electrical energy consumed] $\times$ 100% = 8400000 J / $(1 \times 10^7$ J) * 100% = 84%. The thermal efficiency of the water heater = 84%.
Answer = 84%

---

### 3.2 FORMULA NORMALIZATION

We found that the reasoning steps (i.e. the explanation) in the obtained raw dataset lacked a normalized format and were expressed quite casually. Some formulas mixed parameter names (e.g., "mass of water") and symbols (e.g., "$m_{water}$"), while others simply provided calculations in numerical form without parameter names or symbols. In order to ensure that all explanations adopted a normalized form of formulas, we normalized the formula annotations in the explanations. An example can be found in Table 2. In this process, we need to *identify the formulas used within the original explanations* and to *correct any formatting issues*. Manually undertaking such tasks would require significant effort. However, since the process is not open-ended, but rather structured and verifiable, we could automatically, e.g., *using a LLM*, extract formulas from the explanations, calculate each step, and compare the result with the given answer to ensure the accuracy of this normalization process.

Specifically, to enhance the efficiency of the annotation, we adopted a coarse-to-fine annotation approach with the help of a LLM[2]. We first prompted the LLM in a few-shot manner to generate accurate explanations of the reasoning process. Then, we used few-shot prompts to guide the LLM in correcting minor errors within the normalized explanations, including formatting errors in formula annotations and inaccuracies in the parameters used during computations. Both prompts can be found in Appendix A.1.1. Next, we will provide a detailed description of this process.

Initially, we introduced the question along with its original explanation and the corresponding answer to guide the LLM through few-shot prompting to revise the original explanation. We observed that the ability of the LLM to revise explanations towards normalized explanations remained satisfactory. To assess the correctness of the revised explanations, we extracted formulas from these explanations and then computed the answer using the numbat tool[3]. In addition to providing explanations, we also required the LLM to present the values, symbols, and units of each parameter in the formulas in the form of a table. An example is shown in Figure 1.

At this stage, we checked the correctness of the formula format in the explanations by automatic rules, including whether there were omissions in parameter names, parameter symbols, or corresponding units, and these issues were all correctable. Therefore, if our program detected that the LLM had not successfully generated an accurate normalized explanation, we used few-shot prompting to identify and correct these specific errors. More details can be found in Appendix A.1.1. We observed that the questions which remained incorrect despite multiple attempts by the LLM were of notably poor quality, including missing important reasoning steps, unclear question formulation, and so on. Some examples of these questions can be found in Appendix A.1.2. These questions were removed from our dataset. Following this step, our dataset contains a remaining total of 5,420 questions.

## 3.3 FORMULA DATABASE CONSTRUCTION

Our next step was to *construct a unified formula database for the entire dataset*. Given that parameters in the same formula can be expressed differently across various problem contexts, for instance, the two formulas "[weight of water] = [mass of water] * [gravitational acceleration]" and "[weight] = [mass] * [gravitational acceleration]" both calculate the weight of an object, we need to merge these formulas into a single representation.

Table 3: Changes in the number of formulas after each merging step.

| Step | # Formulas |
|------|------------|
| Before merging | 12,906 |
| After symbolic rules based merging | 1,163 |
| After semantic-based merging | 439 |
| After manual review and error correction | 272 |

We divided the construction process of the formula database into three steps: 1) Merge the formulas through symbolic rules. 2) Merge the formulas through semantic-based method. 3) Manual review and error correction. In Table 3, we present the initial number of formulas and the remaining number of formulas after each step.

**Symbolic rules based merging.** In this step, we merged formulas through symbolic rules. Specifically, this was achieved by *comparing the structure of the formulas and the symbols*. Take the following as an example of judging whether two formulas have the same structure: the formulas "$f_1 : a_1 = (b_1 + c_1)/d_1$", "$f_2 : a_2 = (b_2 + c_2)/d_2$" and "$f_3 : b_1 = a_1 * d_1 - c_1$" have the same structure because $f_2$ can be derived from $f_1$ by renaming parameters, and $f_3$ can be obtained from $f_1$ by transformation. Moreover, in physics, certain physical quantities are conventionally represented by specific symbols. For example, the mass of an object is often denoted by "$m$" and the density of an object is frequently represented by the symbol "$\rho$". Subscripts are then used to distinguish which specific object a physical quantity refers to, such as "$\rho_{water}$" for the density of water. For any two formulas, we first computed all the transformations of each formula to obtain a set of all its variants. Then, we compared the formula structures in the two sets to determine if two formulas were structurally equivalent. If they shared the same structure, we then compared whether their

---

[2]During dataset construction, we accessed Qwen-max via API (https://help.aliyun.com/zh/dashscope/developer-reference/quick-start). Qwen-max is a LLM with over 100B parameters and a strong capability in Chinese.

[3]https://numbat.dev. Numbat is designed for scientific computations with support for physical units.

symbols, with subscripts removed, were identical. If they were, we considered these two formulas to be mergeable. When merging, we retained the parameter with the shorter length from the two. After merging based on symbolic rules, we reduced the number of formulas in the formula database from 12,906 to 1,163.

**Semantic-based merging.** In the symbolic rules based merging process, the semantic information of the parameter names was neglected. This led us to *perform merges grounded on the semantics of the parameter names*. For instance, two formulas that were not merged during the symbolic fusion stage, "[density] = [mass] / [volume]" and "[density of water ] = [mass of water] / [volume of water]", can actually be merged. We would carry out the merging of these two formulas based on the semantic information of the parameter names (for example, "density" and "density of water" are semantically similar). Specifically, for formulas with identical structures, we tokenized each pair of corresponding parameters to create two sets of words[4]. When the two sets overlapped, the parameters were considered to have semantic connection, and the formulas became candidates for merging. Utilizing this approach, we identified a set of pairs of potentially mergeable formulas and then consulted the LLM for a thorough evaluation of each pair. The prompts can be found in Appendix A.1.3. After this step, the number of formulas in the formula database was reduced to 439.

**Manual review and error correction.** Upon completing the aforementioned merging process, we manually inspected the correctness of the results, rectified instances where errors occurred during merging, and manually merged formulas that were overlooked by the LLM. In this process, there were two human volunteers cross-validating the results of manual review and annotation. Finally, we obtained a formula database consisting of 272 formulas.

## 4 EXPERIMENTS SETUP

In this section, we explore several methods for handling the questions within FormulaReasoning, including prompting LLMs using zero-shot and few-shot chain-of-thought (CoT, Wei et al., 2022; Kojima et al., 2022), and training a formula retriever to retrieve formulas to be incorporated into LLM prompts. Additionally, we employed two approaches to enhancing the reasoning abilities of fine-tuned models with fewer than 7B parameters. The first approach involved dividing the reasoning process into distinct steps: formula generation, parameter extraction, and numerical calculation. The second approach leveraged data augmentation to improve the models' reasoning ability.

### 4.1 DATASET SPLIT

We divided FormulaReasoning into into subsets for training, *id* (in-distribution) test, and *ood* (out-of-distribution) test, comprising 4,608, 421 and 391 questions, respectively. We required that all formulas in the id test must appear in the training set, whereas in the ood test, each question involves at least one formula that has not been seen in the training set. This division is designed to evaluate the generalizability of fine-tuned models on formulas that they have not previously encountered.

### 4.2 EVALUATED METHODS

#### 4.2.1 HUMAN PERFORMANCE

We recruited 108 students from a high school, with each student being assigned 7–8 questions. Each student was given 40 minutes to complete these questions. These questions were used as part of their in-class exercises, and at the end, each student received a gift. The final statistics were collected to evaluate human performance, which was consented by all the students.

#### 4.2.2 LLMS

Following Kojima et al., 2022, we incorporated the phrase "Let's think step by step" into the zero-shot prompt to guide LLMs in generating the reasoning steps. For the few-shot setting, we randomly

---

[4]We used jieba: https://github.com/fxsjy/jieba.

sampled five questions from the training set to serve as examples for in-context learning. Each example includes the question text and the reasoning steps (i.e., the explanation). Examples of the prompts can be found in Appendix A.2.2.

We conducted experiments on GPT-4o, GPT-4-turbo, GPT-3.5-turbo, GLM-4-plus, GLM-4-flash (GLM et al., 2024), and Qwen-max. We also evaluated on Qwen2.5-7B/14B (Yang et al., 2024) and Llama3.1-8B (Meta, 2024).

### 4.2.3 Formula Retriever

We trained a formula retriever on the training set. Specifically, we encoded each question using the Chinese-BERT-wwm-base (Devlin et al., 2019; Cui et al., 2021) model to obtain the CLS vector of the question. Each formula in the formula database was represented by a randomly initialized vector. During training, we calculated the cosine score between the question vector and the formula vector. The retriever was then trained with in-batch negatives and contrastive learning loss (Gao et al., 2021). Subsequently, for each question in the id test, we retrieved the top five formulas with the highest scores and included them in the prompt to observe the change in the performance of the LLM when provided with relevant formulas. More details can be found in Appendix A.2.3.

### 4.2.4 Supervised Fine-tuned Models

We found that directly prompting models possessing fewer than 7B parameters failed to produce satisfactory outcomes (for example, ChatGLM3-6B attained merely 8.99 points in a zero-shot setting). Therefore, we conducted supervised fine-tuning of models with fewer than 7B parameters, yet discerned that, dissimilar to larger models (such as GLM-4-plus), smaller models did not exhibit proficient performance in numerical extraction and calculation. In order to augment the reasoning capabilities of smaller models, we explored two approaches for improvement. We conducted experiments on Qwen-1.8B (Bai et al., 2023) and ChatGLM3-6B (Zeng et al., 2022).

**Chain-of-Thought Supervised Fine-Tuning (CoT-SFT)**   We decomposed the reasoning process into several steps. First, we instructed the model to generate the formulas required to solve the question. Subsequently, the parameter names within the formulas were extracted, allowing the model to retrieve the corresponding values and units from the context. Next, the formulas and the associated parameter values were provided to a calculator to obtain the final result. This approach relieved the model from numerical calculation, allowing it to concentrate on the reasoning aspect.

**Data Augmentation (DA)**   We augmented the training dataset with the assistance of larger models. Firstly, we utilized a few-shot approach to prompt a LLM (Qwen-max) to generate new question-answer pairs. The correctness of the computation process generated by the LLM was meticulously verified using a calculator. Subsequently, the formulas generated by the model were extracted and normalized. More details could be found in Appendix A.2.1.

### 4.3 Metric

We utilized numbat to evaluate the predictions generated by the model against the gold-standard answers. A prediction is deemed correct if the relative error (prediction - gold) / gold is less than 1%. We employed *accuracy*, which is the proportion of questions answered correctly, as our metric.

### 4.4 Implementation Details

We accessed to GPT-4o (gpt-4o-2024-08-06 version), GPT-4-turbo (gpt-4-1106-preview version), GPT-3.5-turbo (gpt-3.5-turbo-1106 version)[5], GLM-4-plus, GLM-4-flash[6], Qwen-max and Qwen2.5-7B/14B[7] through API calls with the default hyper-parameters. For Llama3.1, we conducted experiments on NVIDIA V100-32G GPUs. These LLMs generated using nucleus sampling with top_p=0.8. Models that require fine-tuning were experimented on NVIDIA V100 GPUs with

---

[5]https://platform.openai.com/docs

[6]https://open.bigmodel.cn/

[7]https://help.aliyun.com/zh/dashscope/developer-reference/quick-start

Huggingface Transformers and Pytorch 2.0. For Qwen-1.8B, we used a learning rate of 1e-5 and a batch size of 32, and tested the model after training for 10 epochs. For ChatGLM3-6B, we fine-tuned with LoRA (Hu et al., 2021) with r=8, alpha=32 and learning rate of 5e-5, batch size of 1. The max input length and output length are both set to 512. We utilized nucleus sampling with top_p=0.8 for generation. In the case of CoT-SFT, which directly outputted formulas along with corresponding parameter values and units, if the generation output contained formatting errors, we allowed the small model to retry up to 5 times until a correctly formatted output was generated. Training Qwen-1.8B, ChatGLM3-6B models required 12 and 24 hours respectively.

## 5 EXPERIMENTS RESULTS

### 5.1 HUMAN PERFORMANCE

In FormulaReasoning, humans achieved impressive performance, with a score of 93.49 on the id test, 90.47 on the ood test, and an average score of 92.03.

### 5.2 RESULTS OF LLMS

Table 4: Results of LLMs with zero-shot and few-shot prompting.

| Model | Size | zero-shot CoT | | | few-shot CoT | | |
|---|---|---|---|---|---|---|---|
| | | id test | ood test | Avg. | id test | ood test | Avg. |
| GPT-4o | unknown | 77.20 | 72.38 | 74.88 | 76.01 | 73.66 | 74.88 |
| GPT-4-turbo | unknown | 70.07 | 72.89 | 71.43 | 71.50 | 77.49 | 74.38 |
| GPT-3.5-turbo | unknown | 26.13 | 25.58 | 25.87 | 32.07 | 29.92 | 31.03 |
| GLM-4-plus | >100B | **84.32** | **81.07** | **82.76** | **82.90** | **85.68** | **84.24** |
| GLM-4-flash | unknown | 71.50 | 71.87 | 71.68 | 61.76 | 67.01 | 64.29 |
| Qwen-max | >100B | 57.24 | 60.10 | 58.62 | 55.82 | 61.38 | 58.50 |
| Qwen2.5 | 14B | 61.28 | 64.71 | 62.93 | 61.28 | 65.22 | 63.18 |
| Qwen2.5 | 7B | 42.04 | 43.73 | 42.38 | 59.62 | 65.73 | 62.56 |
| Llama3.1 | 8B | 13.06 | 9.74 | 11.46 | 9.74 | 9.72 | 9.73 |
| Human | - | 93.49 | 90.47 | 92.03 | 93.49 | 90.47 | 92.03 |

The evaluation results on LLMs are shown in Table 4. *GLM-4-plus exhibited the best performance in both zero-shot and few-shot settings*, surpassing the second-ranked GPT-4o by an average of 7.88 points in zero-shot setting and 9.36 in few-shot setting. Among models with size not exceeding 20B, Qwen2.5-14B demonstrated commendable performance in both zero-shot and few-shot settings. The subpar performance of Llama3.1 might be due to its pre-training data being primarily in English. After incorporating few-shot examples, GPT-4-turbo, GPT-3.5-turbo, GLM-4-plus and Qwen2.5 demonstrated performance improvements, ranging from 0.25 to 20.18. However, similar performance changes were not observed on other LLMs. Surprisingly, the open-source Qwen2.5-14B model outperformed the closed-source Qwen-max model[8].

Human performance surpassed the performance of the flagship model GLM-4-plus with zero-shot setting and few-shot setting by margins of 9.27 and 7.79 points, respectively. Such results demonstrated that there remained a substantial gap between the current capabilities of state-of-the-art LLMs and human performance. This was even more pronounced when considering smaller-scale models. These findings underscored *the challenging nature of FormulaReasoning as an unresolved dataset*, and that there was significant room for improvement in LLMs as they struggled to match human levels of reasoning.

We also compared the chain of thought (CoT) and program of thought (PoT, Chen et al., 2023) methods, with the results presented in Appendix A.2.4. The results indicated that CoT and PoT demonstrated varying performances between different models and under different settings.

---

[8]We have not yet found clear information indicating whether the closed-source Qwen-max is also based on version 2.5.

## 5.3 RESULTS OF LLMS WITH FORMULA RETRIEVER

The results of LLMs utilizing the formula retriever are shown in Table 5. We found that the impact on performance varied among different LLMs when incorporating retrieved formulas into prompts. We observed a positive enhancement on Qwen2.5-7B, with score increments of 10.92 and 4.04 with zero-shot and few-shot, respectively, on the id test. However, we found that the performance was essentially on par on the GLM-4-flash. Specifically, we found that the top 5 retrieved formulas often included irrelevant ones, as the number of formulas required varies for each problem. The presence of these extraneous formulas affected the model's performance, indicating that there is considerable room for further research in retrieving from a formula database.

Table 5: Results of LLMs with Formula Retriever on the id test.

| Model | zero-shot | few-shot |
|---|---|---|
| GLM-4-flash | 71.50 | 61.76 |
|   + formula retriever | 70.55 | 62.95 |
| Qwen2.5-7B | 42.04 | 59.62 |
|   + formula retriever | 52.96 | 63.66 |

## 5.4 RESULTS OF SUPERVISED FINE-TUNED MODELS

Table 6 shows the results for the supervised fine-tuned models, with and without CoT-SFT and DA, which were detailed in Section 4.2.4. In most settings, both models achieved higher scores on the id test than the ood test, yet they still exhibited considerable performance on the ood test. This indicates that *1) the ood formulas indeed challenged model performance* and *2) the models still demonstrated a certain level of generalizability.* We hope that the division of id test and ood test will be helpful for assessing the generalization ability of fine-tuned models in future work.

Table 6: Results of supervised fine-tuned models on FormulaReasoning.

| Model | Size | id test | ood test | Avg. |
|---|---|---|---|---|
| Qwen-1.8B | | 55.91 | 44.58 | 50.25 |
|   + DA | 1.8B | 56.16 | 45.32 | 50.74 |
|   + CoT-SFT | | 73.65 | 74.38 | 74.00 |
| ChatGLM3-6B | | 52.95 | 40.64 | 47.02 |
|   + DA | 6B | 53.44 | 45.32 | 49.53 |
|   + CoT-SFT | | 74.63 | 73.89 | 74.23 |

It was noteworthy that with CoT-SFT, Qwen-1.8B and ChatGLM3-6B, with a mere parameter count of 1.8B and 6B, respectively, achieved performance comparable to GPT-4o (though such a comparison may not be entirely fair). This indicated that the incorporation of CoT-SFT and the use of calculators could significantly enhance the reasoning capabilities of small models. Our findings revealed that *focusing on reasoning with CoT while delegating numerical calculation to a calculator could enhance the performance of small models*, given their limited calculating capability. *The assistance of LLMs for data augmentation could also enhance smaller models' reasoning capability.* This finding provides valuable insights for future deployment of numerical reasoning systems with small models.

## 5.5 CASE STUDY AND ERROR ANALYSIS

We sampled 50 error cases from the id test (few-shot setting) of GPT-3.5-turbo and manually categorized the types and proportions of errors. We divided the error types into two main categories: *formula errors* and *calculation errors*. Formula errors encompass inappropriate formulas and omitted formulas, while calculation errors primarily involve inaccuracies in numerical calculation and unit errors. We found that 38% of errors were caused by incorrect formulas, while the remaining 62% were attributable to calculation errors. We provide one example for each of the two types of errors listed in Figure 2. It could be observed that FormulaReasoning poses challenges to existing models in terms of formula application and numerical calculation (including unit calculation and arithmetic calculation).

**Question**

The maximum power of a tank engine is 7.2×10^5W. During a certain mission, the tank traveled 3000m on a straight road at a constant speed using its maximum power for 300 seconds, consuming 15kg of diesel fuel. (q_diesel = 4.0×10^7J/kg) Q: The efficiency of the tank's engine.

**Output of GPT-3.5-turbo** (wrong formula highlighted in red)

Solution: 1. Calculate the time it takes for the tank to travel: … *omitted* …
3. Calculate the engine efficiency: [Engine efficiency] = [Engine output power] / [Heat energy consumed by fuel] … *omitted* …

**Gold Explanation** (correct formula highlighted in green)

Solution: 1. Calculate the heat released by the complete combustion of diesel fuel: … *omitted* …
3. Calculate the efficiency of the tank's engine: [Tank engine efficiency] = [Work done by the engine] / [Heat released by complete combustion of diesel fuel] × 100% Equation … *omitted* …

(a) An error case caused by wrong formulas.

**Question**

Given that the calorific value of natural gas is 3.8×10^7 J/m^3 and the specific heat capacity of water is 4.2×10^3 J/(kg·°C). If the efficiency of a natural gas stove in heating water is 50%, then at standard atmospheric pressure, how many degrees Celsius can 10L of natural gas, burning completely, heat 3.0kg of water initially at 60°C?

**Output of GPT-3.5-turbo** (wrong calculation highlighted in red. Note that 1 L = 10^{-3} m^3)

Solution: 1. Calculate the heat content of natural gas: [Heat content of natural gas] = [Calorific value of natural gas] × [Volume of natural gas] Calculation = 3.8×10^7 J/m^3 × 10 L = 3.8×10^8 J Heat content of natural gas = 3.8×10^8 J … *omitted* …

**Gold Explanation** (correct calculation highlighted in green)

Solution: 1. Calculate the heat released from the complete burning of natural gas:
[Heat released from complete combustion of natural gas] = [Volume of natural gas] × [Calorific value of natural gas]
Calculation = 10 L × 3.8×10^7 J/m³ = 3.8 × 10^5 J … *omitted* …

(b) An error case caused by wrong calculation.

Figure 2: Error cases.

## 6 CONCLUSION AND LIMITATIONS

We introduced FormulaReasoning, a dataset for formula-based numerical reasoning. We annotated the reasoning steps with formulas for each question with both manual and LLM-assisted efforts. Furthermore, we constructed a formula database after merging formulas with similar meanings, serving as an external knowledge base for subsequent retrieval-based/augmented approaches. We evaluated FormulaReasoning across various sizes of LLMs, supervised fine-tuned models, and retrieval-augmented LLMs, demonstrating its challenging nature as an unresolved task. Our findings indicate substantial room for improvement of existing models on formula-based numerical reasoning, thus motivating future research efforts.

In the future work, we plan to utilize the formula knowledge from FormulaReasoning to improve the numerical reasoning capabilities of LLMs. Possible approaches include enhancing reasoning abilities through knowledge-driven methods, preference learning methods based on formula feedback.

One limitation of this work is that our evaluation results reported in the paper were obtained from the original Chinese version of FormulaReasoning. We have employed a combination of LLM-based translation and manual review to release an English version of FormulaReasoning. Currently, we provide a preview English version in our GitHub repository, and we will release the official English version of FormulaReasoning after completing the manual review process. Another limitation is that, our dataset is limited to the domain of physics. Although junior high school physics is not overly complex and can be understood by most people which would benefit evaluation efforts, it is still possible to explore formula-based question answering data in other domains.

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

# A  APPENDIX

## A.1  DATASET CONSTRUCTION

### A.1.1  PROMPTS IN FORMULA NORMALIZATION

The process of formula normalization is delineated into three distinct stages: the generation of natural language explanations, the extraction of the associated parameters from the explanations, and the subsequent error correction phase. The initial two stages are illustrated in Figures 3 and 4. The third stage is further splited into three specific error categories, each addressed by a dedicated prompt: input errors, where the parameters mentioned in the explanation are absent from the question; calculation errors, which occur when the calculator reports an error during the computation process; and output errors, where the final computed answer is incorrect. We provide an example here focusing on prompts for correcting calculation errors, while prompts for the other two error types can be found in our code submission. The prompts designed to correct calculation errors are depicted in Figure 5. The entire normalization procedure employs a 6-shot prompting, an instance of which is provided herein for illustrative purposes.

### A.1.2  EXAMPLES OF DELETED QUESTIONS

The questions which remained incorrect despite multiple attempts by the LLM were of notably poor quality, including missing important reasoning steps, wrong reference answer, and so on. Here is an example of these questions in Figure 6.

### A.1.3  SEMANTIC-BASED MERGING FOR FORMULA DATABASE CONSTRUCTION

Semantic-based merging primarily employs the LLM to comprehend formulas, ascertain if two formulas are semantically equivalent, and subsequently determine whether they can be merged into a single formula. The prompt for this procedure is illustrated in Figure 7. This approach ensures that the nuanced meanings embedded within formulas are accurately captured and evaluated for potential merging, thereby enhancing the quality of formula database.

## A.2 EXPERIMENTS

### A.2.1 DATA AUGMENTATION (DA) FOR FORMULAREASONING

There have been several studies utilizing large language models (LLMs) for data augmentation (Ding et al., 2024). The data generated in these related works (Zheng et al., 2023; Whitehouse et al., 2023) primarily focus on daily conversations or sentiment analysis and do not require rigorous numerical calculations. Some research on data augmentation involving numerical calculations (Shum et al., 2023) employs LLMs to generate solutions to questions to aid in training, rather than creating complete questions. In contrast to these approaches, our work generates complete questions that involve numerical calculations (particularly formula calculations), along with automatic improvement and selection to ensure data quality.

In order to enhance the capabilities of models, we use LLM to generate more data for fine-tuning. We divide the process of data generation into the following several steps.

First, we randomly generated 17,000 prompts. Each prompt was obtained by stacking five question-answer pairs sampled form training set. At the end of the prompt, LLM was required to generate the sixth question-answer pair. Second, we normalized the generated formulas. Except for the absence of manual review, the remaining steps were consistent with those in Section 3.2. At last, we unitized the calculator to check whether the calculation process in the data generated by the LLM is correct, and discarded the generated data with incorrect calculation processes. After the above steps, we finally retained more than 2500 questions.

We found that mixing the newly generated data into the original training set did not always bring positive improvement, perhaps because the newly generated data has not undergone manual review. We found that randomly selecting a small portion of the newly generated data can enable the model to have performance improvement. We set several different mixing ratios selected from $\{5\%, 10\%, 15\%, 20\%, 2\%, 30\%, 35\%, 40\%\}$. We fine-tuned each model using the augmented data set. After training for a fixed number of steps (150k and 200k), we selected the checkpoints with the smallest loss among models of different mixing ratios.

### A.2.2 ZERO-SHOT AND FEW-SHOT PROMPTS

Zero-shot and few-shot prompts are shown in Figure 8.

### A.2.3 FORMULA RETRIEVER

Let the number of formulas in the formula database be $N$. During training, we randomly initialized a matrix $\mathbf{F} \in \mathbb{R}^{N \times d}$, where $d$ is the hidden size and the $i$-th row in $\mathbf{F}$ represented the initial representation of the $i$-th formula in formula database. We denoted a batch of questions with a batch size of $B$ as $Q = \{q_1, q_2, ..., q_B\}$. The indices of the gold-standard formulas corresponding to these $B$ questions were denoted as $L = \{l_1, l_2, \cdots, l_B\}$ (i.e. the label of $q_i$ is $l_i$, where $1 \leq i \leq B$).

BERT was utilized to encode each question,

$$\mathbf{h}_{cls}^i, \mathbf{h}_1^i, \cdots = \texttt{BERT}(q_i), 1 \leq i \leq B. \tag{1}$$

Subsequently, we took the CLS vector $\mathbf{h}_{cls}^i$ as the representation for the $i$-th question.

We utilized in-batch negatives and contrastive learning loss,

$$\mathcal{L} = -\frac{1}{B} \sum_{1 \leq i \leq B} \log \frac{\exp(\cos(\mathbf{h}_{cls}^i, \mathbf{F}_{l_i}))}{\sum_{1 \leq j \leq B} \exp(\cos(\mathbf{h}_{cls}^i, \mathbf{F}_{l_j}))}. \tag{2}$$

Each question might correspond to multiple correct formulas, and we ensured that the same question did not appear twice in the same batch when loading the data. Based on the implementation of Chinese-BERT-wwm-base, we tested the retrieval performance on the id test set and found that Recall@5 reached 97.69%.

Models were evaluated with top-5 retrieved formulas. Prompts can be found in Appendix A.2.5. We utilized zero-shot CoTs.

Table 7: Results of LLMs with zero-shot and few-shot chain of thought (CoT) and program of thought (PoT).

| Model | zero-shot | | | few-shot | | |
|---|---|---|---|---|---|---|
| | id test | ood test | Avg. | id test | ood test | Avg. |
| GPT-4o (CoT) | 77.20 | 72.38 | 74.88 | 76.01 | 73.66 | 74.88 |
| GPT-4o (PoT) | 80.76 | 73.91 | 77.46 | 81.47 | 82.61 | 82.02 |
| GLM-4-plus (CoT) | 84.32 | 81.07 | 82.76 | 82.90 | 85.68 | 84.24 |
| GLM-4-plus (PoT) | 84.08 | 78.51 | 81.40 | 86.70 | 84.91 | 85.84 |
| Human | 93.49 | 90.47 | 92.03 | 93.49 | 90.47 | 92.03 |

### A.2.4 COMPARISON OF COT AND POT PROMPTS

Results are shown in Table 7. In the PoT approach, we utilized a Python interpreter to execute the code and obtain the final results. We found that the performance comparison between CoT and PoT varies across models. GPT-4o consistently demonstrated superior performance with PoT across all settings, achieving improvements of 2.58 points on average in the zero-shot setting and 7.14 points on average in the few-shot setting. In contrast, GLM-4-plus showed an average decline of 1.36 points in the zero-shot setting but showed an average improvement of 1.60 points in the few-shot setting. The finding might be related to the code capabilities of the models.

### A.2.5 PROMPTS FOR LLMS WITH FORMULA RETRIEVER

We added the formulas before each question in the few-shot setting. For the examples sampled from the training set, gold-standard formulas were added before each question. For the final question from the test set in both zero-shot and few-shot prompts, we included the top 5 retrieved formulas. The prompts are shown in Figure 9.

| **Prompt actually used** | **English translation** |
|---|---|
| 我需要你修改问题原有的解析，给出规范格式的新解析，要求如下：
1.请逐步地进行思考,如果有公式组合的部分需要一步步地拆分成基本公式进行求解
2.公式中的计算符号，如"+"、"-"、"×"、"/"和"^"不能省略
3.公式需要同时给出符号和有具体含义的两种形式，然后代入数值计算得出答案
4.涉及到单位换算的部分需要展示出来具体过程
5.使用"[]"标注公式中的变量，其中科学计数法形式的数字"a×10^b"以及复杂的单位，需要使用"()"标注
6.如果有latex格式的公式，比如"\frac{Q_吸}{Q_放}"需要改成正常算式的形式："Q_吸/Q_放"
7.解析开始如果给出"（解析可能存在错误）"表明给出的解析中可能存在错误
下面是一些例子：
input:
问题：在某次测试中该车匀速行驶时受到的阻力是车重的 0.02 倍，在 0.5h 内行驶了 28km 的路程，求：行驶的平均速度为多少？
解析：解：车辆在0.5h内行驶了 28km 的路程，所以行驶的平均速度为：V=s/t=28km/0.5h=56km/h。
output:
解：
原始解析比较清晰，下面将其转换为规范格式：
第一步，先计算平均速度：
v=s/t
[平均速度]=[路程]/[时间]
算式=28km/0.5h=56km/h
平均速度=56km/h
答案=56km/h
***...omitted...***
下面是一个新的问题：
问题：**{{question}}**
解析：**{{explanation}}** | I need you to modify the original explanation of the question and provide a new explanation with the following requirements:
1. Please think step by step. If there has formula combination, you need to decompose the combination into basic formulas step by step.
2. Calculation symbols such as "+", "-", "×", "/" and "^" in formulas cannot be omitted.
3. The formula needs to be given in both symbolic and concrete forms. After that, you need to substitute into the numerical calculation to obtain the answer.
4. The part related to unit conversion needs to show the specific process.
5. Use "[]" to label variables in formula, "()" for numbers like "a×10^b" in scientific notation and for complex units.
6. If you have a latex formula such as \frac{Q_in}{Q_out}, change it to a normal formula: "Q_in /Q_out".
7. Start with "(Explanation may be wrong)" indicates that there may be an error in the given explanation.
Here are some examples:
input:
Question: In a test, the resistance of the car at a constant speed is 0.02 times the weight of the car, and the car traveled a distance of 28km in 0.5h. What is the average speed?
Explanation: The car has traveled a distance of 28km in 0.5h.The average speed of the vehicle is V=s/t=28km/0.5h=56km/h.
output:
Solution:
The raw parsing is clear, so let's convert it to a canonical format:
1.we compute the average speed:
v=s/t
[Average speed]=[distance]/[time]
expression=28km/0.5h=56km/h
Average speed =56km/h
Answer=56km/h
***...omitted...***
There's a new question:
Question: **{{question}}**
Explanation: **{{explanation}}** |

Figure 3: Prompt of the formula normalization stage 1.

| **Prompt actually used** | **English translation** |
|---|---|
| 我需要你帮我从给定的问题和解析中提取出解决问题使用到的公式，以及公式对应的参数表：
1.请逐步地进行思考,先对解析进行分析，然后生成提取结果，如果有公式组合的部分需要一步地拆分成基本公式进行求解
2.公式中所有的计算符号，如"+"、"-"、"×"、"/"和"^"不能省略
3.公式中的每个变量需要使用"[]"标注出来，而且变量需要使用有意义的文字标识，尽量避免直接使用数值
4.如果有latex格式的公式，比如"\frac{Q_吸}{Q_放}"需要改成正常算式的形式："[Q_吸] / [Q_放]"，算式中的单位换算部分不属于公式，不需要被提取
5.参数表中的参数是公式中使用到的参数（参数名称要与公式中的参数一致），表格包括：概念、符号、数值、单位,使用"|"分割单元格
6.参数表的数值和单位来自于问题本身以及解析计算的中间结果，如果参数进行了单位换算，参数表要给出原始的参数形式（没有进行单位换算）
7.参数表中的参数如果没有单位或者是表示某个物体的数量，比如"3盏灯"、"转了8圈"、"4个人"等等，那么单位写入"null"
8.提取结果中给出的公式和参数表中不需要任何额外的解释信息
下面是一些例子：
input:
问题: 在某次测试中该车匀速行驶时受到的阻力是车重的 0.02 倍，在 0.5h 内行驶了 28km 的路程，求：行驶的平均速度为多少?
解析: 解:
原始解析比较清晰，下面将其转换为规范格式:
第一步，先计算平均速度:
v=s/t
[平均速度]=[路程]/[时间]
算式=28km/0.5h=56km/h
平均速度=56km/h
答案=56km/h
output:
提取分析:
1.解析中的公式格式符合要求，计算符号没有省略，变量使用"[]"标注，而且变量使用有意义的文字标识
2.计算过程中没有进行单位换算，参数可以直接提取
下面给出提取结果（单位换算部分不属于公式，不需要被提取；参数表中不需要额外解释信息）
提取结果:
1.先计算平均速度，根据路程和时间的关系:
[平均速度]=[路程]/[时间]
参数表:
概念\| 符号\| 数值\| 单位
路程\| s\| 28\| km
时间\| t\| 0.5\| h
平均速度\| v\| 56\| km/h
*...omitted...*
下面是一个新的问题:
问题: **{{question}}**
解析: **{{explanation}}** | I need you to help me extract the formula and parameters table for the formula from given question and explanation:
1. Think step by step, analyze the explanation first, and then generate the extracted results. If there is a combination of formulas, the combination needs to be split into basic formulas step by step.
2. All calculation symbols such as "+", "-", "×", "/" and "^" in the formula cannot be omitted.
3. Each variable in the formula needs to be labeled with "[]", and the variable needs to be identified with meaningful text instead of numbers.
4. If a latex formula such as \frac{Q_in}{Q_out} needs to be changed to a normal formula: [Q_in]/[Q_out]. The unit conversion does not need to be extracted.
5. The parameters table come from the parameters in formula (the parameter name should be consistent with the parameters in formula), the table include: concept, symbol, numeric, unit, using cell division "\|".
6. The numeric and unit in the parameter table come from the problem itself and the intermediate results of analytical calculation. If the parameters are converted into different units, the parameter table should give the original parameter form (without unit conversion).
7. If the parameter in the parameters table has no units or represents the amount of an object, such as "3 lights", "8 revolutions", "4 people", etc., then the units are written as "null".
8. Apart from formula and parameter table, no additional information is required in the extraction results.
Here are some examples:
input:
Question: In a test, the resistance of the car at a constant speed is 0.02 times the weight of the car, and the car traveled a distance of 28km in 0.5h. What is the average speed?
Explanation:
1.we compute the average speed:
v=s/t
[Average speed]=[distance]/[time]
expression=28km/0.5h=56km/h
Average speed =56km/h
Answer=56km/h
output:
Extraction analysis:
1. The formula format in the analysis meets the requirements. The calculation symbols are not omitted. Variables are labeled with "[]", and variables are expressed with meaningful text.
2. No unit conversion was performed during the computation, and parameters can be directly extracted.
Below is the extraction result (the unit conversion part does not belong to the formula and does not need to be extracted; no additional explanatory information is required in the parameter table).
Extraction result:
1. First calculate the average speed, based on the relationship between distance and time:
[average speed]=[distance]/[time]
Parameter table:
Concept \| Symbol \| Numeric \| Unit
distance \| s \| 28 \| km
time \| t \| 0.5 \| h
average speed \| v \| 56 \| km/h
*...omitted...*
There's a new question:
Question: **{{question}}**
Explanation: **{{explanation}}** |

Figure 4: Prompt of the formula normalization stage 2.

| **Prompt actually used** | **English translation** |
|---|---|
| 我需要你帮助我纠正解析中的错误，我会给出问题和错误信息，下面是错误纠正的要求：
1.你需要先进行错误分析，分析如何修改来纠正错误，然后给出错误纠正部分，纠正解析中的错误
2.错误纠正部分不需要任何额外解释信息，错误纠正部分的格式为："内容：修改前的内容->修改后的内容"，增加内容时"修改前的内容"为null，删除内容时"修改后的内容"为null
3.问题缺失参数：如果问题中没有缺失的参数，那么向题目中增加缺失的参数;如果问题中的参数与缺失参数的含义相同但格式不同，修改题目中的参数与缺失参数相同
4.算式错误：算式存在错误需要对公式和错误的参数进行修改，如果算式中存在"[参数]"或"null"，需要补齐缺失的参数；如果参数没有问题可能需要对公式进行修改
5.公式的格式为"[待求解参数]=[参数1](+\|-\|×\|/)[参数2]..."；参数表的格式为："概念\|符号\|数值\|单位"，比如"水的沸点是100℃，表示为"水的沸点\| t_沸\| 100\| ℃"

下面是一些例子：
input:
问题：假设13.0t烟煤在煤炉中完全燃烧，放出的热量部分被水吸收，可以使4×10^5kg的水从20℃升高到100℃，求水吸收的热量为多少J [c_水=4.2×10^3J / （kg·℃)]
错误信息：
算式错误：1.计算水升高的温度差：
公式：[水升高的温度差]=[末温]-[初温]
算式=[末温]-[初温]
问题缺失参数：水升高的温度差=80 ℃；
output:
错误分析：
1.根据错误错误信息：算式存在错误，而且算式中存在"[参数]"的情况："[末温]"、"[初温]"，需要对参数表增加缺失的参数
根据错误错误信息，"[末温]-[初温]"，从题目中可以找到相关文本"从20℃升高到100℃"，按照要求的参数格式表示为：
初温\| t_0\| 20\| ℃
末温\| t\| 100\| ℃
这样参数表增加缺失的参数后，代入1.计算水升高的温度差的公式可以得到：
算式=((100) ℃)-((20) ℃)=80 ℃
水升高的温度差=80 ℃
2.根据错误信息，问题缺失参数，由于分析1中纠正算式后计算得到了"水升高的温度差=80 ℃"，所以问题不再缺失参数，不需要进行修改

错误纠正：
参数表：null->初温\| t_0\| 20\| ℃
参数表：null->末温\| t\| 100\| ℃
*...omitted...*
下面是一个新的问题：
问题：**{{question}}**
错误：**{{error}}** | I need your help to correct the error in the explanation. I will provide the question and error information. The following are the requirements for error correction:
1. You need to first conduct error analysis, analyze how to modify to correct the error, and then provide the error correction to correct the error in the explanation.
2. The error correction section does not require any additional explanatory information. The format of the error correction section is: "Content: Pre modified Content ->Modified Content". When adding content, "Pre modified Content" is null, and when deleting content, "Modified Content" is null.
3. Missing parameters in the question: If there are no missing parameters in the question, add the missing parameters to the question; If the parameters in the question have the same meaning as the missing parameters but different formats, modify the parameters in the question to be the same as the missing parameters.
4. Expression error: The formula and incorrect parameters need to be modified. If there is "[parameter]" or "null" in the expression, the missing parameters need to be filled in; If there are no issues with the parameters, it may be necessary to modify the formula.
5. The format of the formula is "[parameter to be solved]=[parameter 1] (+\|- \|×\|/) [Parameter 2]… "; The format of the parameter table is: "concept \| symbol \| numeric \| unit", for example, "The boiling point of water is 100 ℃", which is represented as "the boiling point of water \| t_boiling \| 100 \| ℃"
Here are some examples:
input:
question: Assuming that 13.0 tons of bituminous coal is completely burned in a coal furnace, the heat released is partially absorbed by water, which can make 4×10^5kg of water is raised from 20 ℃ to 100 ℃, how much heat is absorbed by the water ? [c_water=4.2×10^3J/(kg · ℃)]
Error message:
Expression error: 1. Calculate the temperature difference of water rise: Formula: [temperature difference of water rise]=[final temperature] - [initial temperature]
expression=[final temperature] - [initial temperature]
Question missing parameter: Temperature difference of water rise=80 ℃;
output:
Error analysis:
1. According to the error message: there is an error in the formula, and there are cases of "[parameter]" in the formula: "[final temperature]" and "[initial temperature]". It is necessary to add them to the parameter table.
According to the error message, "[Last Temperature] - [Initial Temperature]", the relevant text "Increase from 20 ℃ to 100 ℃" can be found in the title. According to the required parameter format, it is represented as:
Initial temperature \| t_ 0 \| 20 \| ℃
Final temperature \| t \| 100 \| ℃
After adding missing parameters to the parameter table, substitute them into the formula for calculating the temperature difference can be obtained as follows:
expression=((100) ℃)-((20) ℃)=80 ℃
temperature difference of water rise=80 ℃
2. According to the error message, the question is missing parameters. After analyzing the correction equation in 1 step, it was calculated that "the temperature difference of water rise=80 ℃", so the question is no longer missing parameters and does not need to be modified.
Error correction:
Parameter table: null ->Initial temperature \| t_ 0 \| 20 \| ℃
Parameter table: null ->final temperature \| t \| 100 \| ℃
*...omitted...*
There's a new question:
Question: **{{question}}**
Error: **{{error}}** |

Figure 5: Prompt of the formula normalization stage 3: error correction for "calculation error".

**Question:**
As shown in the figure, the Xuelong 2 scientific research icebreaker designed in China.
***...omitted...*** When traveling at a constant speed of 3.6km/h in thick ice covered waters, the resistance experienced by the icebreaker is approximately $2×10°N$. Calculate the propulsion power of the icebreaker at this time.
Reference answer: $2×10^7$ W

**Formula:**
[thrust]=[resistance]
[propulsion power]=[thrust]×[constant speed]

Parameter table:

| Parameter | symbol | value | unit |
|-----------|--------|-------|------|
| resistance | f | $2×10^7$ | N |
| ship speed | v | 1 | m/s |

**Explanation:**
1.Calculate thrust:
thrust=resistance=$2×10^7$N
2.Calculate propulsion power:
propulsion power=thrust×constant speed=$2×10^7$N×constant speed *(cannot find value)*

***Error:***
1. *The parameter "resistance" in the question is in the incorrect format.*
2. *"constant speed" could not be located in the parameter table.*

Figure 6: An example of deleted questions.

| **Prompt actually used** | **English translation** |
|--------------------------|-------------------------|
| 下面我会给出两个公式，每个公式由参数和运算符号构成，[]中的表示参数。
你需要判断我给出的两个公式中对应参数表达含义是否相同，是否是同一个公式：
如果含义不相同，不是同一个公式，只需要回答不是；
如果各个参数含义相同，是同一个公式，则需要给出最终的公式，并且给出一个三行的表格来表示参数的对应关系，每个单元格内容是一个参数，前两行填写两个公式的参数，第三行填写统一后的公式参数。
下面是公式1：
{**公式 1**}
下面是公式2：
{**公式 2**}
通过表达含义判断，是否是同一个公式： | I will give two formulas below. Each formula consists of parameters and operation symbols. The text in [] represent parameter.
You need to judge whether the corresponding parameters in the two formulas I gave have the same meaning and whether they are the same formula:
If the meaning is different, and they are not the same formula, just answer no;
If each pair of parameters have the same meaning, and they are the same formula, the final formula needs to be given, and a three-row table needs to be given to indicate the corresponding relationship between the parameters. The content of each cell is a parameter, and the first two rows are filled with two formulas. Parameters, fill in the unified formula parameters in the third row.
Here is formula 1:
{**formula 1**}
Here is formula 2:
{**formula 2**}
Judge whether they are the same formula by their meanings: |

Figure 7: Prompt for semantic-based merging.

| Prompt actually used | English translation |
|---|---|
| 这是一个初中物理题目，根据问题给出计算的过程，让我们一步一步地思考，在最后用"###"作为开始给出最终答案（一个数字）和答案的单位。

Question: {{**问题**}}
Answer: | This is a junior high school physics question. Based on the given question, provide the calculation process and let's think step by step. Finally, use "###" to start giving the final answer (a number) and the unit of the answer.

Question: {{**question**}}
Answer: |

(a) Zero-shot prompt for LLMs.

| Prompt actually used | English translation |
|---|---|
| 这是一个初中物理题目，根据问题给出计算的过程，用公式表示。

Question: {{**样例1问题**}}
Answer: {{**样例1解析**}}

*…omitted…*

Question: {{**问题**}}
Answer: | This is a junior high school physics question. Based on the given question, provide the calculation process.

Question: {{**question of example 1**}}
Answer: {{**explanation of example 1**}}

*…omitted…*

Question: {{**question**}}
Answer: |

(b) Few-shot prompt for LLMs.

Figure 8: Zero-shot and few-shot prompts for LLMs.

| Prompt actually used | English translation |
|---|---|
| 这是一个初中物理题目，根据问题给出计算的过程，用公式表示。

可能用到的公式有: {{**top 5检索到的公式**}}
Question: {{**问题**}}
Answer: | This is a junior high school physics question. Based on the given question, provide the calculation process.

The formulas that may be used include: {{**top 5 retrieved formulas**}}
Question: {{**question**}}
Answer: |

(a) Few-shot prompt for LLMs with formula retriever.

| Prompt actually used | English translation |
|---|---|
| 这是一个初中物理题目，根据问题给出计算的过程，用公式表示。

可能用到的公式有: {{**用到的公式**}}
Question: {{**样例1问题**}}
Answer: {{**样例1解析**}}

*…omitted…*

可能用到的公式有: {{**top 5检索到的公式**}}
Question: {{**问题**}}
Answer: | This is a junior high school physics question. Based on the given question, provide the calculation process.

The formulas that may be used include: {{**used formulas**}}
Question: {{**question of example 1**}}
Answer: {{**explanation of example 1**}}

*…omitted…*

The formulas that may be used include: {{**top 5 retrieved formulas**}}
Question: {{**question**}}
Answer: |

(b) Zero-shot prompt for LLMs with formula retriever.

Figure 9: Zero-shot and few-shot prompts for LLMs with formula retriever.