# OpenReview forum: "FormulaReasoning: A Dataset for Formula-Based Numerical Reasoning"
_ICLR.cc/2025/Conference — ICLR 2025 Conference Withdrawn Submission_

### Official Review · Reviewer_pbC8 · 2024-10-29

**Soundness:** 3
**Presentation:** 4
**Contribution:** 2
**Rating:** 3
**Confidence:** 4

**Summary:**

The paper introduces a dataset of 5,420 questions to test formula-based reasoning of LLMs. The main contribution of this dataset is that

1. All questions require use of formulas.
2. Dataset contains a larger number of formulas (272) compared to existing datasets such as Math23K-F and MAWPS-F.

The questions are parsed from China high school physics examinations so I believe they rather test the ability to apply physics formulas rather than mathematics. The authors apply some manual and LLM based preprocessing steps in order to get a clean dataset where ever question is accompanied by a formula.

The authors test existing LLMs on this benchmark and demonstrate that LLMs still perform worse than humans on this task (%84 vs %92).

**Strengths:**

FormulaReasoning introduces a unique dataset that has a particular focus on formula-based reasonings. The authors test a wide-range of LLMs on this dataset and compare to the performance of high school students on these problems.

The paper clearly explains the methodology on how the dataset was collected and how the benchmarking was done. Furthermore, it is very well written and easy to follow.

**Weaknesses:**

The paper's main technical contribution is that it introduces a *formula-based* reasoning dataset, however, I believe that there are existing datasets such as GPQA (https://arxiv.org/abs/2311.12022) and MMLU (https://arxiv.org/abs/2009.03300) that has questions that require the use of formulas. This renders the contribution of this paper relatively limited.

The analysis of where the existing models fail is also limited - authors show two examples of failures but do not further elaborate on failure modes or statistics. In addition, I think presenting the failures of the strongest model rather than GPT-3.5 would be more informative.

The reason I gave a 3 is that the existing benchmarks already require formula-based reasoning - so the main/only benefit of this paper seems to me that the formulas are explicitly written out. However, I don't see how this helps with evaluation of the models. It could be helpful if these formulas were novel ones that do not appear in model's training data, in which case, the formula could be included in the prompt to test model's ability to include it (or the model could be asked to retrieve it in a "RAG" style). However, most of these formulas are still mainstream ones that the model would see in its training. With this in mind, it seems to me that this dataset is more of an additive set to existing benchmark and lack technical novelty required for ICLR publications.

**Questions:**

- Did you scan MMLU and GPQA diamond for formula based reasoning? I suspect these datasets must have formula based reasoning similar to your dataset rather than GSM8K, Math23K-F, or MAWPS-F.
- Did you try testing GPT-o1's capabilities on your benchmark? It would be valuable to see how much improvement that results in.
- Can you provide more examples where the top-performing model fails? I believe it's important to understand when and why these models fail. Is it mostly calculation or formula mistakes?

---

> ### Author Response · Authors · 2024-11-15
>
> Many thanks for your review.
>
> **Regarding the role of formulas in the dataset** We aim for FormulaReasoning to enhance the reasoning abilities of LLMs through explicitly annotated formulas. In fact, in Section 5.3, we used a RAG-based approach to enhance the reasoning abilities of LLMs.
>
> **Q1**: Yes, we have also reviewed the MMLU and GPQA datasets. These datasets do not focus solely on numerical reasoning but include other types of reasoning as well, which is why we did not directly compare them with FormulaReasoning in our paper. However, the numerical reasoning questions in these datasets do not have formula annotations, which aligns with the GSM8K dataset that we directly compare with.
>
> **Q2**: We will add the evaluation results of the o1 model to our paper.
>
> **Q3**: Thanks for your suggestion. We will add more error analysis.

---

> > ### Comment · Reviewer_pbC8 · 2024-11-26
> >
> > Thanks for the clarifications. In light of other reviews, most prominently the one with o1 evaluation and assessment of the difficulty of the benchmark, I will keep my score unchanged.

---

### Official Review · Reviewer_MMuW · 2024-11-03

**Soundness:** 3
**Presentation:** 2
**Contribution:** 2
**Rating:** 3
**Confidence:** 4

**Summary:**

The paper presents FormulaReasoning, a new numerical reasoning dataset comprising 5,420 questions with reasoning paths and corresponding formulas. The dataset's raw questions are sourced from Chinese junior high school physics exams, then filtered and annotated through a collaborative process involving human annotators and LLMs. Experiments benchmark various LLMs and prompting methods, including zero-shot, few-shot, and retrieval-augmented approaches.

**Strengths:**

- The data collection process is meticulous (e.g., formula normalization and the construction of formula databases), resulting in a high-quality reasoning dataset with annotated formulas.
- The experiments provide a solid baseline for the benchmark, covering a range of LLM approaches effectively.

**Weaknesses:**

- The motivation for creating this dataset lacks clarity. In the introduction, the authors state that “Current datasets … do not reflect the complexity of real-world problems.” However, it appears that formulas are neither a necessary nor a sufficient condition for solving complex numerical problems. Similarly, on line 41, the dataset is described as requiring “domain-specific formulas to guide the numerical reasoning process,” yet this new requirement does not necessarily demonstrate that formulas add value. From my perspective, annotated formulas are particularly valuable for reducing hallucinations and enhancing interpretability, especially when LLMs are applied in scientific domains where this matters more. The authors may wish to clarify the significance of their dataset for the broader community.
- The metric does not include formula validation. According to Section 4.3, the metric only measures the accuracy of the final result. It raises the question of why annotated formulas are introduced if LLMs can produce correct answers without explicitly displaying formulas.
- The dataset’s coverage is limited. It consists solely of physics questions (5,420 samples) with 272 unique formulas. As a retrieval database or a testbed for LLM formula knowledge, expanding to more domains and increasing the dataset size would enhance its applicability.

**Questions:**

The example in Figure 1 is presented in English, despite references to Chinese-specific techniques (e.g., using Chinese-BERT-wwm-base as the retriever). Clarification on the dataset's current language and any future plans for language expansion would be helpful.

---

> ### Author Response · Authors · 2024-11-15
>
> Thank you for your review.
>
> **W1**: We agree with your perspective that "annotated formulas are particularly valuable for reducing hallucinations and enhancing interpretability, especially when LLMs are applied in scientific domains where this matters more.". We will revise the relevant descriptions in the paper accordingly.
>
> **W2**: Thank you for your suggestion. However, existing numerical reasoning evaluations primarily focus on the final result, as the correct intermediate steps are not unique and are difficult to evaluate. We will address how to evaluate intermediate results in future work.
>
> **W3**: Thank you for your suggestion. However, we would like to clarify that the number of 272 formulas and the number of formulas in fields such as physics after merging are limited.

---

> > ### Comment · Reviewer_MMuW · 2024-11-20
> > **Reviewer response**
> >
> > Thank you for your response and the clarifications provided. Sorry, I will need to keep my score unchanged at this time. I believe the paper would be better suited for the conference after a more thorough re-positioning to align with its core contributions.
> >
> > Regarding the third point, I understand the constraints due to the limited availability of formulas in physics. However, I believe the work would be significantly strengthened by expanding its scope to include more scientific domains beyond physics, thereby broadening its applicability and impact.

---

### Official Review · Reviewer_Y6yk · 2024-11-03

**Soundness:** 3
**Presentation:** 3
**Contribution:** 1
**Rating:** 3
**Confidence:** 3

**Summary:**

The key contributions of the paper are
- a dataset consisting of 5420 reasoning-based questions from Chinese junior high school physics examinations, where each question requires the use of formulas to solve.
- a formula database containing 272 unique formulas used in the above questions, that were assembled by a meticulous process of symbolic,  LLM-based, and manual filtering.

The authors also evaluate a range of LLMs on a subset of their dataset.

**Strengths:**

The dataset construction process is well-documented and thorough. I've looked into the supplementary files at https://anonymous.4open.science/r/FormulaReasoning, and I commend the authors for enabling the reviewers to inspect the dataset easily.

This is, to the best of my knowledge, the first dataset with such an extensive categorization of the mathematical formulas used in every problem.

Although the test split is too small, the overall dataset itself large enough to reliably detect small improvements in accuracy.

**Weaknesses:**

The paper says, verbatim:
> Such results demonstrated that there remained a substantial gap between the current capabilities of state-of-the-art LLMs and human performance. This was even more pronounced when considering smaller-scale models. These findings underscored _the challenging nature of FormulaReasoning as an unresolved dataset_, and that there was significant room for improvement in LLMs as they struggled to match human levels of reasoning.

I prompted _o1-preview_ with the problems from `id_test.json` https://anonymous.4open.science/r/FormulaReasoning/id_test.json in the supplementary material. The dataset seems to be in no particular order, so to avoid cherry-picking, I took the first 20 problems in the dataset. The prompt was "Translate the problem to English, then solve it. question: [problem in Chinese]".

The score was 19/20. Looking into the only one it missed:
```
  "如图所示，是平底热水壶，其质量为0.8kg，内底面积为180cm^2．某次用该热水壶装1.2L水放在水平桌面上，测得水深10cm，初温25℃。[ρ_水=1.0*10^3kg/m^3，C_水=4.2*10^3J/（kg℃），g=10N/kg]加热前水对壶底的压力多大？",
```
translates to:
```
As shown in the figure, this is a flat-bottomed kettle with a mass of 0.8kg and an inner bottom area of ​​180cm^2. Once, the kettle was filled with 1.2L of water and placed on a horizontal table. The water depth was measured to be 10cm and the initial temperature was 25℃. [ρ_water=1.0*10^3kg/m^3, C_water=4.2*10^3J/(kg℃), g=10N/kg] How much pressure does the water exert on the bottom of the kettle before heating?
```
The ground truth answer is given as 18 N, while the model gives the answer in units of Pa. However, the dataset lacks figures, so it's unclear if all necessary information is provided. Additionally, the correct answer seems odd, as the question asks for pressure, but N is a unit of force. This discrepancy could be due to a misunderstanding of the original Chinese statement by both myself and _o1-preview_.


I suspect a better prompting + a cleaned dataset would make even _o1-mini_ score very close to 100% when run on the entire dataset.


Summary:
- I strongly disagree with the characterization of  “the challenging nature of FormulaReasoning as an unresolved dataset”. The dataset is very easy for today's LLMs and is not useful as a benchmark.
- The paper does not present a clear vision for a use of this dataset apart from benchmarking LLMs;
- Apart from the _o1_ series, the paper lacks evaluations for Claude-3.5-Sonnet and Llama-3-405b;
- While the dataset filtering and formula identification methods are quite involved, the overall utility of this dataset remains unclear. Specifically, the purpose of the identified formulas is not well-defined. A "formula database" derived from high school physics questions may be an interesting artifact for educational research, but its relevance to this conference is less apparent.

**Questions:**

1. Could you evaluate smarter models on your benchmark?
2. What are the identified formulas useful for, exactly?
3. What do you see as the default use case of this dataset in the next year?

---

> ### Author Response · Authors · 2024-11-15
>
> Many thanks for your review.
>
> **Regarding the performance on the o1 model** We will revise the relevant descriptions in the paper. The o1 model is a highly capable but computationally expensive and closed-source model. However, many other LLMs fail to achieve satisfactory results on FormulaReasoning, as evidenced by the evaluation results presented in our paper. The o1 model achieves a score of 94.8 on the math competitions level MATH dataset (see [OpenAI reference](https://openai.com/index/learning-to-reason-with-llms/)). Similarly, early versions of GPT-4 achieved 92% accuracy on GSM8k (see [GPT-4 paper](https://cdn.openai.com/papers/gpt-4.pdf)). Nonetheless, this does not preclude datasets like MATH and GSM8k from continuing to serve as representative benchmarks for evaluating numerical reasoning in LLMs.
> We will also provide evaluation results on the o1, Claude-3.5-Sonnet and Llama-3-405b.
>
> **Regarding the role of formulas in the dataset and the formula database** We integrate the formula database with the RAG approach in our paper to enhance the reasoning capabilities of LLMs (see Section 5.3). We hope that a reasoning dataset where each step is guided by formulas can inspire further research on topics such as knowledge-augmented reasoning and step-by-step supervised training methods.

---

> > ### Comment · Reviewer_Y6yk · 2024-11-23
> >
> > Thank you for the comment. I maintain my score. One of the main selling points of the submission was a difficult benchmark.
> > In the absence of the difficulty claim, I think the work is basically correct, but do not agree with the formula database being enough of a scientific contribution to the machine learning community to be accepted at ICLR.

---

### Official Review · Reviewer_bpwS · 2024-11-05

**Soundness:** 3
**Presentation:** 2
**Contribution:** 3
**Rating:** 5
**Confidence:** 4

**Summary:**

This paper presents a novel question-answering dataset called FormulaReasoning for evaluating formula-based numerical reasoning capabilities of language models. The dataset addresses a critical gap in numerical reasoning benchmarks by explicitly incorporating formulas and domain-specific knowledge. The paper thoroughly describes the dataset construction process and presents baseline results using various state-of-the-art language models and fine-tuning approaches. The FormulaReasoning dataset contains 5,420 questions covering 272 formulas.

**Strengths:**

* *Novel dataset addressing a critical gap*: The FormulaReasoning dataset fills a clear need for benchmarks that evaluate formula-based reasoning, going beyond simple arithmetic to test domain-specific knowledge applications.
* *Rigorous dataset construction process*: The multi-step process for collecting, annotating, and normalizing the questions and formulas is well-described and appears to be carefully designed.
* *Comprehensive evaluations*: The experiments cover a wide range of model types and sizes, providing a thorough picture of current capabilities. The results highlight the challenges of formula-based reasoning for current models and demonstrate the potential of techniques like Chain-of-Thought supervised fine-tuning for improving performance.

**Weaknesses:**

1. The idea of *Formula Retriever* does not make sense. Why will the representation of the question and the required formula be similar? For a given question, let's say z = x/y is the right formula to use.  The representation of z = x*y and z = x/y will be very similar. Unless this is handled (maybe using special tokens), the model will always struggle to choose the right formula.
2. While there is a brief error analysis, a more in-depth examination of the types of mistakes made by different models could provide additional insights.
3. While the focus on physics is well-motivated, discussing how the approach might generalize to other formula-heavy domains would strengthen the paper.

**Questions:**

1. What is the use of a *unified formula database*? What if we don't merge the formulas? What happens?
2. How does performance change if there are more variables in the formula? For example z = x+y (2 variables) and z = a*b + c/d (4 variables). Do models struggle if there are many variables in the formula?

---

> ### Author Response · Authors · 2024-11-15
>
> Thank you for your detailed review.
>
> **W1**: You may have some misunderstandings. Your concerns are valid, but this issue will not arise in our formula retriever. As mentioned in Section 4.2.3, 'Each formula in the formula database was represented by a randomly initialized vector.' We do not directly use an encoder to encode the formula; instead, the representation of the formula is randomly initialized and updated through training, allowing the model to learn the intrinsic meaning of the formula, which remains consistent regardless of formula transformations.
>
> **Q1**: As mentioned in Section 3.3, the same formula may have different representations across various questions. Therefore, when constructing the formula database, it is necessary to standardize these formulas for the subsequent use of formula database for LLMs, such as the RAG method.
>
> **Q2**: Currently, FormulaReasoning addresses the multivariable formulas by breaking down the reasoning steps, a method that is equally applicable to formulas with multiple variables.
>
> We appreciate the reviewer’s suggestions and will include more error analysis and give more analysis in how to generalize to other formula-heavy domains.

---

> > ### Comment · Reviewer_bpwS · 2024-11-28
> >
> > Thank you for your detailed clarification. My score remains unchanged.

---

### Note · Authors · 2025-01-15

**Comment:**

We decide to withdraw this submission. Thank you.

**Withdrawal Confirmation:**

I have read and agree with the venue's withdrawal policy on behalf of myself and my co-authors.